

# Outcomes of metabolic syndrome and anxiety levels in light and heavy smokers

Musa Şahin and Didem Yüzügüllü

Provincial Health Directorate of Adana, Ministry of Health Türkiye, Adana, Turkey

## ABSTRACT

**Background:** This study aimed to assess the impact of smoking status, as measured by pack-years (PY), on components of metabolic syndrome while considering the influence of anxiety.

**Design:** This cross-sectional study was conducted at a smoking cessation clinic in Turkey, enrolling individuals who visited the clinic in 2022. The Fagerstrom Test for Nicotine Dependence and the State-Trait Anxiety Inventory were utilized as assessment tools, while metabolic syndrome parameters (body mass index, hypertension, hyperglycemia, dyslipidemia) were evaluated. Smoking status was classified based on pack-years.

**Results:** The study revealed a dose-dependent relationship between smoking status and essential metabolic factors such as systolic blood pressure (SBP), diastolic blood pressure (DBP), hemoglobin A1c (HbA1c), and low-density lipoprotein (LDL). Notably, triglyceride (TG) levels exhibited a significant increase, particularly at 25 pack years. While anxiety levels did not exhibit a significant correlation with smoking status, they demonstrated an upward trend with increasing SBP and DBP values. Anxiety levels did not exhibit a significant correlation with smoking status.

**Conclusions:** A significant association was identified between nicotine addiction, as indicated by PY, and both metabolic syndrome parameters and anxiety levels. Early smoking cessation is strongly recommended for current smokers, and former smokers are advised to abstain from smoking to mitigate its adverse effects on metabolic syndrome components. These findings underscore the interconnectedness of cigarette smoking's effects on both physical and mental health, emphasizing the necessity of comprehensive approaches encompassing both metabolic disorder management and mental health support within cessation programs.

## INTRODUCTION

Smoking constitutes a chronic addiction to nicotine, acknowledged as a significant risk factor for a spectrum of diseases and disabilities impacting both physical and mental well-being (*Pawlina et al., 2015*). Tobacco smoking is one of the biggest public health risks in the general population. According to the World Health Organization (WHO), over 8 million people die annually due to tobacco-related causes, with over 7 million directly attributable to tobacco use, and 1.2 million fatalities resulting from health complications associated with passive smoking (*WHO, 2022*). If current smoking trends persist,

Corresponding author
Musa Şahin,
dr.musasahin@hotmail.com

projections suggest that more over 8 million individuals will succumb annually to tobacco-related illnesses by the year 2030 (*WHO, 2011*). Given the conclusive evidence of nicotine's harm in health contexts, the latest International Classification of Diseases (ICD-11) introduces substantial improvments, significant updates and perspectives on nicotine-related disorders (*World Health Organisation*). Notably, individuals with psychiatric disorders exhibit a heightened prevalence of nicotine addiction, as nicotine's psychoactive properties often temporarily alleviate psychiatric symptoms to some extent (*El-Sherbiny & Elsary, 2022*). Despite mounting evidence of its detrimental effects, smoking remains ingrained in the belief system of many, who perceive it as a stress reliever and anxiety-reducer (*CDC, 2013*). Particularly noteworthy is the disproportionately elevated smoking prevalence among individuals with psychiatric conditions such as schizophrenia, anxiety disorders, eating disorders, and attention deficit hyperactivity disorder (*Sciberras et al., 2017*).

A literature review highlights a significant failure rate in smoking cessation efforts. Studies examining smoking cessation programs underscore a concerning trend of high failure rates, largely attributed to elevated levels of depression, anxiety, and stress, coupled with a general reluctance to adopt behavioral changes (*Ghali et al., 2019*; *Shepherd et al., 2022*). Anxiety, given its role as a potential motivator for smoking initiation, emerges as a crucial focus area warranting investigation (*McDermott et al., 2013*). Notably, smokers grappling with heightened anxiety levels often experience poorer outcomes in cessation efforts, marked by intensified withdrawal symptoms and a heightened urge to smoke (*Johnson et al., 2019*; *Lewis et al., 2020*). Moreover, apprehensions surrounding nicotine withdrawal symptoms and heightened state anxiety further compound the challenges of quitting smoking for individuals with elevated anxiety levels (*McLaughlin, Dani & De Biasi, 2015*). On a positive note, emerging evidence suggests smoking cessation can complement treatment strategies for mental disorders, particularly anxiety disorders, yielding favorable effects on mental health outcomes (*Cavazos-Rehg et al., 2014*; *Taylor et al., 2014*).

Metabolic syndrome, characterized by a cluster of cardiovascular risk factors, including abdominal obesity, hypertension, hyperglycemia, and dyslipidemia, represents a burgeoning public health concern with implications for diabetes and cardiovascular morbidity; approximately one-quarter of the global adult population is documented to have metabolic syndrome (*American Diabetes Association, 2013*; *Wang et al., 2022*). Lifestyle shifts toward sedentary behaviors and high-calorie diets have fueled the escalating prevalence of metabolic syndrome, posing a formidable threat to recent strides in health advancement (*Saklayen, 2018*; *Cena et al., 2013*).

The association between anxiety and smoking behavior often leads individuals to smoke more, thereby increasing their exposure to the adverse effects of smoking and the risk of developing metabolic syndrome. This underscores the importance of conducting comprehensive evaluations and providing integrated services for individuals seeking smoking cessation assistance at outpatient clinics. There is a need in the literature in terms of studies which concurrently investigate smoke consumption, anxiety, and metabolic syndrome. Additionally, studies on the effect of nicotine addiction on metabolic syndrome

have yielded conflicting results, highlighting the need for further research. Therefore, this study aimed to evaluate the effect of smoking status, nicotine dependence on metabolic syndrome parameters and simultaneously examining anxiety levels in smokers. By examining these factors concurrently, we aim to contribute to a better understanding of the complex interplay between smoking behavior, psychological factors, and metabolic health, thereby informing more effective strategies for smoking cessation interventions and metabolic syndrome management.

## MATERIALS AND METHODS

### Sampling

This retrospective, cross-sectional investigation was carried out at a smoking cessation outpatient facility within the provincial health directorate of Adana, located in Southern Turkey, targeting individuals expressing a willingness to cease smoking. Enrollment comprised only current smokers seeking assistance at the smoking cessation outpatient clinic, with inclusion spanning all visitors to the facility throughout 2022 during their initial consultation. Laboratory tests and Fagerstrom nicotine assessments are routinely performed, as usual. Anxiety tests were conducted for all clients during the study year. All information is recorded electronically and in a physical file. At the end of the study year, sample selection was conducted through simple randomization of the records in the file. For individuals with an overlooked medical test deficiency, another participant was substituted through simple randomization.

All visitors who applied to the outpatient clinic for smoking cessation between 01.01.2022 and 31.12.2022 were included in the study. The data of the study were formed with the evaluations of the clients at their first application to the outpatient clinic. Inclusion criteria included being a smoker (having a history of smoking regularly every day), having applied to a smoking cessation outpatient clinic between 01.01.2022 and 31.12.2022, being over 18 years of age and under 65 years of age, having complete examination and interview information, being literate, volunteering to quit smoking and agreeing to participate in the study. People who reported smoking at least one cigarette a day and who were still smoking were considered smokers. Those who used any tobacco product such as cigarettes, cigars, hookahs, pipes were included in the study and referred to as "smokers" in this study. After an interview in an outpatient clinic serving clients who volunteered to quit smoking five days a week, those who were decided to be smokers, whose treatment process was initiated and who met the inclusion criteria were included in the study. During this initial visit, comprehensive medical evaluations—including blood tests, medical history, and family history—were conducted and subsequently filed for each individual. Files were later reviewed for data collection in this study. Exclusion criteria were predetermined confounding factors, including a current body mass index exceeding 30, presence of coronary artery disease, diabetes, receipt of antihypertensive/ antihyperlipidemic/antihyperglycemic treatment, diagnosed psychiatric illness, and individuals falling outside the age range of 18 to 65 years.

This study is subject to potential selection biases due to its inclusion and exclusion criteria. By excluding individuals with chronic conditions such as coronary artery disease,

diabetes, or a BMI over 30, the study focused on a healthier subset of smokers. Additionally, the study population was limited to individuals actively seeking assistance at a smoking cessation outpatient clinic, which may differ systematically from the general smoking population in terms of motivation and readiness to quit smoking. As such, these factors should be considered when interpreting the findings of this study.

Inclusion, exclusion criteria, and sample size determination paralleled a comparable study conducted by *Cena et al. (2013)*. Sample size analysis was executed employing the Epi Info Sample size calculator (www.openepi.com), predicated on a metabolic syndrome prevalence rate of 52%, with a statistical power of 97%, a confidence interval of 95%, and a design effect of 1.0, yielding a sample size of 397 participants drawn from a pool of 2,052 individuals seeking smoking cessation assistance at the clinic in 2022 (*Saklayen, 2018*). Among the study population of 397 participants, 52.4% (208 individuals) met the predefined inclusion criteria.

## Blood tests and measurements

Adana Public Health Laboratory works on the tests requested by the physicians of Family Health Centers (FHC) and Community Health Centers (TSM), which are primary healthcare institutions. All routine biochemistry, hormone, hemogram and hemoglobin A1c (HbA1c) samples of smokers who were evaluated with blood tests in the smoking cessation outpatient clinic between the dates covering the study period were retrospectively analyzed.

HbA1c analysis was performed using the Bio-Rad Variant II Turbo device (Bio-Rad Laboratories, Hercules, CA, USA) according to the manufacturer's instructions. The measurement principle was based on cation exchange chromatography. Lipid levels (triglycerides, HDL, and LDL cholesterol) and glucose were measured using dedicated kits on a biochemical autoanalyzer (Siemens ADVIA 2400 Chemistry Analyzer; Siemens Healthcare GmbH, Erlangen, Germany).

Height and weight measurements of current smokers who apply to the smoking cessation outpatient clinic are conducted during their initial visit to compute body mass index (BMI). Subsequently, biochemical parameters, including hemoglobin A1c levels, insulin, fasting blood glucose (FBG) concentrations, low-density lipoprotein (LDL), high-density lipoprotein (HDL), total cholesterol, triglyceride (TG) levels, and blood pressure (BP) readings were documented. Insulin resistance is quantified utilizing the Homeostatic Model Assessment of Insulin Resistance (HOMA-IR) formula: [FBG (mg/dL) × Fasting insulin (μU/mL)]/405. Elevated blood pressure, in accordance with guidelines, is defined as readings surpassing 130/85 mmHg and categorized as "high" BP (*Williams et al., 2018*).

## Nicotine dependence evaluation

The Fagerstrom Nicotine Dependence Test is always utilized to measure the nicotine dependence levels of smokers seeking assistance at the smoking cessation outpatient clinic. Developed by Karl O. Fagerstrom, this test determines the level of physical dependence on cigarettes and consists of six closed-ended questions (*Fagerstrom & Schneider, 1989*).

This instrument comprises six closed-ended questions designed to gauge the degree of physical dependence on cigarettes, and scores on the test increase with higher levels of dependence. Individuals scoring below five points are classified as mild nicotine addicts, those scoring between five and six points as moderate nicotine addicts, and those scoring seven points or above as severe nicotine addicts. The Turkish validity and reliability study of the test was conducted by *Uysal et al. (2004)*, revealing moderate reliability ($\alpha = 0.56$).

## Anxiety assessment

In the study, the State-Trait Anxiety Inventory (STAI) developed by Spielberger is employed to assess the participants' anxiety levels (*Spielberger & Vagg, 1984*). The STAI is a scientifically validated self-report questionnaire comprising 40 items, designed to differentiate between state anxiety, which is temporary, and trait anxiety, which reflects a more general tendency to experience anxiety across various situations. This inventory encompasses two sections: the STAI-1, measuring state anxiety, and the STAI-2, assessing trait anxiety. The scale comprises two parts: STAI-1, which evaluates state anxiety levels, and STAI-2, which assesses trait anxiety levels. It underwent adaptation into Turkish, with validity and reliability studies conducted between 1974 and 1977 (*Öner & LeCompte, 1998*). The State-Trait Anxiety Inventory consists of two separate scales with a total of twenty items each. In the State-Trait Anxiety Inventory, there are ten inverted statements. These are items 1, 2, 5, 8, 10, 11, 15, 16, 19 and 20. In the Trait Anxiety Scale, the number of reversed statements is seven. These are items 1, 6, 7, 10, 13, 16, 19. The total score of direct and reversed statements is calculated. The total score obtained for the direct statements is subtracted from the total score obtained for the reversed statements (*Öner & LeCompte, 1998*; *Spielberger & Vagg, 1984*). According to this scale, three groups were defined for the STAI-S and STAI-T according to the literature: below 37 was considered normal, 37 to 48 was considered moderate, and above 48 was considered severe, high score (*Livadas et al., 2011*). Light to moderate anxiety groups were amalgamated, focusing on severe anxiety.

## Smoking status

Smoking status was classified using pack years (PY), employing three distinct categorizations 1—initially, the study's median PY of 25.0 delineated two groups: light and heavy smokers. Subsequently, in alignment with prior research, smokers were categorized into three groups based on their smoking history. 2—secondly; the first group comprised light smokers with up to 20 pack yearpack years (PY), the second group consisted of moderate smokers with a history of 20 to 39 PY, and the third group included heavy smokers with 40 or more PY (*Li et al., 2011*). 3—third; an alternative approach involved categorizing smokers into 10-year quartiles based on their smoking habits. The first (1st) quartile represented smokers with a history of up to 10 PY, the second (2nd) quartile included smokers with a history of 10 to 20 PY, the third (3rd) quartile encompassed smokers with a history of 20 to 29 years PY and the fourth (4th) quartile comprised smokers with a history of 30 years or more PY (*Shin, Oh & Cho, 2018*). PY was calculated by multiplying the number of cigarettes smoked per day by the number of years of smoking and dividing the result by 20 (*Forey, Thornton & Lee, 2011*).

## Statistical analysis

Normality was assessed using skewness and kurtosis values, with thresholds of −1 to +1 indicating normal distribution. Group comparisons were performed using one-way ANOVA, with Tukey's test for *post-hoc* analysis when variances were equal (Levene's test $p \geq 0.05$) or the Games-Howell test otherwise. For normally distributed variables, Pearson's correlation coefficient and Student's t-test were used, while Spearman's rank correlation and the Mann-Whitney U test were applied for non-normally distributed data. Multivariate analysis was conducted to identify factors influencing nicotine dependence. Results were reported as mean ± standard deviation (SD) or median (IQR), with $p < 0.05$ considered significant. Statistical analyses were performed using SPSS version 24.0.

Ethical clearance for the study was obtained from Çukurova University on 08.04.2022, with acceptance number 22, and informed consent was obtained from all participants (Ethical Application: 121/22, Date: 08.04.2022). The study adhered to the ethical principles outlined in the 1964 Declaration of Helsinki and its subsequent revisions.

## RESULTS

In Table 1, heavy smoking was represented by the median, with all subjects having smoked for a median duration of 25.0 years. Table 1 also outlines the characteristics of the sample and compares values between genders. The data comprises 208 patients, with 97 males and 111 females. The average age of the entire sample was 43.8 ± 11.8 years, while the mean age upon starting smoking was 12.5 ± 9.8 years, with an average smoking duration of 23.3 ± 10.7 years. Basal glucose levels averaged 96.0 ± 18.6 mg/dL. STAI-A and STAI-T Anxiety scores were recorded as 49.4 ± 6.6 and 50.0 ± 7.0, respectively. When categorizing smoking status into heavy and light based on median values, 53.7% of participants were identified as heavy smokers. Prevalence rates of meeting metabolic syndrome criteria were 41.8% for TG, 33.7% for HDL, 23.6% for BP, and 16.4% for FG.

Females had significantly lower HDL levels compared to males ($p < 0.001$). Females exhibited higher systolic blood pressure ($p < 0.001$) but lower diastolic blood pressure ($p < 0.001$). The proportion of heavy smokers was significantly higher among males ($p = 0.008$). Males also had a higher prevalence of elevated triglycerides ($p = 0.001$) and elevated blood pressure ($p = 0.006$) compared to females.

Table 2 categorized values based on light, moderate, and heavy smoking levels by pack years, further divided into quartiles of 10 years each. Age increases significantly with smoking intensity. Light smokers are younger on average (35.4 years), while heavy smokers are older (53.3 years). A similar trend is seen in the quartiles, with the 1st quartile being the youngest (31.3 years) and the 4th quartile the oldest (50.2 years). Significant age differences were noted across the smoking intensity categories and within the 3rd and 4th quartiles ($p < 0.05$).

In the quartile analysis, only the 1st quartile (the lightest smokers) shows a statistically significant longer delay in smoking their first cigarette, with an average of 19.0 min. This delay is significantly longer than that observed in the other quartiles ($p < 0.05$).

The duration of smoking years impacted the smoking level across all quartiles. LDL levels increase with smoking intensity, showing a significant difference between light

**Table 1  Baseline demographics and key characteristics of the sample population[*].**

| Variables | | Men\nn = 97 | Women\nn = 111 | Total\nn = 208 | p |
|---|---|---|---|---|---|
| Age (years) | | 43.1 ± 11.9 | 44.5 ± 11.7 | 43.8 ± 11.8 | 0.414 |
| First cigarette (min) | | 11.9 ± 9.7 | 13.0 ± 10.0 | 12.5 ± 9.8 | 0.421 |
| Years smoked (years) | | 24.2 ± 11.7 | 22.4 ± 9.7 | 23.3 ± 10.7 | 0.233 |
| Basal glycemia (mg/dL) | | 96.2 ± 17.5 | 95.8 ± 19.5 | 96.0 ± 18.6 | 0.870 |
| TG (mg/dL) | | 176.1 ± 88.3 | 141.9 ± 85.0 | 157.9 ± 88.0 | 0.005 |
| HDL Col (mg/dL) | | 44.5 ± 10.2 | 54.4 ± 17.6 | 49.8 ± 15.4 | 0.000 |
| LDL Col (mg/dL) | | 127.1 ± 35.5 | 125.8 ± 36.3 | 126.4 ± 35.8 | 0.797 |
| Insulin | | 13.3 ± 12.9 | 11.1 ± 6.4 | 12.1 ± 10.0 | 0.138 |
| HbA1c | | 5.6 ± 0.6 | 5.7 ± 0.9 | 5.7 ± 0.8 | 0.232 |
| SBP (mm/hg) | | 112.3 ± 15.0 | 114.0 ± 13.7 | 117.9 ± 14.8 | 0.000 |
| DBP (mm/hg) | | 77.7 ± 12.0 | 71.8 ± 10.5 | 74.5 ± 11.6 | 0.000 |
| BMI | | 26.8 ± 4.1 | 26.4 ± 5.5 | 26.6 ± 4.9 | 0.569 |
| State anxiety | | 50.1 ± 7.0 | 48.8 ± 6.1 | 49.4 ± 6.6 | 0.135 |
| Trait anxiety | | 50.7 ± 7.3 | 49.4 ± 6.6 | 50.0 ± 7.0 | 0.160 |
| Smoking status by median[a] | Light | 46 (39.0) | 72 (61.0) | 118 (46.3) | 0.008 |
| | Heavy | 51 (56.7) | 39 (43.3) | 90 (53.7) | |
| Triglyceride | Normal | 45 (37.2) | 76 (62.8) | 121 (58.2) | 0.001 |
| | Elevated | 52 (59.8) | 35 (40.2) | 87 (41.8) | |
| HDL[b] | Normal | 66 (49.6) | 67 (50.4) | 133 (66.3) | 0.157 |
| | Low | 31 (41.3) | 44 (58.7) | 75 (33.7) | |
| Blood pressure[c] | Normal | 66 (41.5) | 93 (58.5) | 159 (76.4) | 0.006 |
| | Elevated | 31 (63.3) | 18 (36.7) | 49 (23.6) | |
| Fasting glucose | Normal | 80 (46.0) | 94 (54.0) | 174 (83.6) | 0.403 |
| | Elevated | 17 (50.0) | 17 (50.0) | 34 (16.4) | |
| State anxiety | Normal | 40 (40.4) | 59 (59.6) | 99 (47.6) | 0.096 |
| | High | 57 (52.3) | (47.7) | 109 (52.4) | |
| Trait anxiety | Normal | 40 (40.8) | 58 (59.2) | 98 (47.1) | 0.127 |
| | High | 57 (51.8) | 53 (48.2) | 110 (52.9) | |
| Addiction levels | Normal | 18 (42.9) | 24 (57.1) | 42 (20.2) | 0.608 |
| | High | 79 (47.6) | 87 (52.4) | 166 (79.8) | |

**Notes:**
[*] Mann-Whitney U and Student t-tests were used for continuous data based on the distribution type. For frequency data, the chi-square test was used. The percentages of counts are represented with row percentages in the "Female" and "Male" columns, while column percentages are used in the "Total" column.
[a] The sample was divided into two groups using all participants' median pack-years (25.0).
[b] HDL values were evaluated separately according to the standards for females and males.
[c] Individuals with blood pressure equal to or above 130/85 were determined to have elevated blood pressure.

smokers (113.6 mg/dL) and moderate/heavy smokers (133.3, 130.8, $p < 0.05$). A similar trend appears across quartiles, with the 1st quartile having the lowest LDL (101.1, $p < 0.05$). HbA1c increases from light to heavy smokers, with a statistically significant difference between light smokers (5.5) and moderate/heavier groups (5.8, 5.9, $p < 0.05$). SBP is significantly higher in heavy smokers (125.0 mm Hg) compared to light and moderate smokers (113.0, 118.1, $p < 0.05$). DBP also increases with smoking intensity, with heavy

**Table 2 Subgroup analysis of baseline characteristics in light, moderate, and heavy smokers[*].**

| Variable | Smoking habits by pack years | | | Quartiles by pack years | | | |
|---|---|---|---|---|---|---|---|
| | Light (0–19) (n = 67) | Moderate (20–39) (n = 97) | Heavy (40–) (n = 44) | 1st quartile (low) (n = 21) | 2nd quartile (n = 46) | 3rd quartile (n = 59) | 4th quartile (n = 82) |
| Age (years) | 35.4 ± 10.1[*] | 45.4 ± 9.8[*] | 53.3 ± 9.6[*] | 31.3 ± 8.8 | 37.2 ± 10.2 | 44.7 ± 10.2[*] | 50.2 ± 9.9[*] |
| First cigarette (min) | 14.2 ± 11.0 | 12.2 ± 9.7 | 10.4 ± 7.8 | 19.0 ± 12.9[*] | 12.1 ± 9.4 | 12.0 ± 9.7 | 11.4 ± 8.8 |
| Years smoked (years) | 14.0 ± 7.9[*] | 24.3 ± 6.4[*] | 35.2 ± 8.9[*] | 8.2 ± 6.8[*] | 16.6 ± 6.9[*] | 22.5 ± 6.0[*] | 31.4 ± 8.7[*] |
| Basal glycemia (mg/dL) | 93.0 ± 11.5 | 97.4 ± 19.3 | 97.4 ± 22.3 | 91.9 ± 13.1 | 93.5 ± 14.9 | 98.0 ± 20.3 | 97.0 ± 20.3 |
| TG (mg/dL) | 135.8 ± 77.8 | 165.8 ± 86.9 | 174.1 ± 99.9 | 142.7 ± 103.4 | 132.6 ± 64.1 | 163.9 ± 90.9 | 171.6 ± 91.3 |
| HDL Col (mg/dL) | 49.4 ± 11.2 | 50.6 ± 18.9 | 48.5 ± 12.2 | 45.5 ± 11.8 | 51.2 ± 10.5 | 52.3 ± 22.2 | 48.3 ± 12.0 |
| LDL Col (mg/dL) | 113.6 ± 31.4[*] | 133.3 ± 36.8 | 130.8 ± 35.5 | 101.1 ± 30.9[*] | 119.3 ± 30.3 | 128.8 ± 35.2 | 135.1 ± 37.1 |
| Insulin | 11.5 ± 11.9 | 12.4 ± 7.2 | 12.6 ± 12.1 | 10.1 ± 5.9 | 12.1 ± 13.8 | 11.9 ± 6.4 | 12.9 ± 10.5 |
| HOMA-IR | 2.9 ± 4.1 | 3.2 ± 2.6 | 3.4 ± 4.7 | 2.4 ± 1.9 | 3.1 ± 4.8 | 3.1 ± 2.5 | 3.4 ± 3.9 |
| HbA1c | 5.5 ± 0.6[*] | 5.8 ± 0.7 | 5.9 ± 1.1 | 5.4 ± 0.7 | 5.5 ± 0.5 | 5.8 ± 0.7 | 5.8 ± 1.0 |
| SBP (mm/hg) | 113.0 ± 12.0 | 118.1 ± 13.1 | 125.0 ± 19.1[*] | 112.6 ± 11.2 | 113.1 ± 12.4 | 117.8 ± 14.3 | 122.0 ± 16.2 |
| DBP (mm/hg) | 71.2 ± 10.7 | 75.3 ± 10.9 | 78.0 ± 13.2[*] | 71.1 ± 10.1 | 71.3 ± 11.0 | 74.7 ± 11.2 | 77.1 ± 12.0[*] |
| BMI | 25.8 ± 6.0 | 27.2 ± 4.5 | 26.3 ± 3.6 | 24.5 ± 5.2 | 26.4 ± 6.2 | 27.0 ± 4.6 | 27.0 ± 4.0 |
| State anxiety | 48.1 ± 5.7 | 49.8 ± 7.5 | 50.7 ± 5.6 | 48.1 ± 7.1 | 48.0 ± 4.9 | 49.6 ± 7.5[*] | 50.4 ± 6.5 |
| Trait anxiety | 48.3 ± 5.9 | 50.5 ± 7.5 | 51.6 ± 6.9[*] | 46.9 ± 6.4 | 49.0 ± 5.6 | 50.4 ± 8.3 | 51.2 ± 6.5 |

**Note:**
[*] Tukey's test $p < 0.05$ for comparisons between variables.

smokers having significantly higher DBP (78.0 mm Hg) than light and moderate smokers (71.2, 75.3, $p < 0.05$).

Heavy smokers have a significantly higher Trait Anxiety score (51.6) compared to light smokers (48.3) and moderate smokers (50.5), with a significant difference observed ($p < 0.05$). Significant differences in State Anxiety scores are observed in the 3rd quartile of smoking intensity, with $p < 0.05$.

In Table 3, Significantly lower mean age ($p = 0.012$) and duration of the first cigarette ($p < 0.001$) were noted when comparing normal to moderate addiction levels with high levels. Significantly lower mean age ($p = 0.012$) and shorter duration until the first cigarette ($p < 0.001$) were noted when comparing normal to moderate addiction levels with high levels. However, in multivariate analysis, age was not significant (OR = 1.0, $p = 0.299$), while the duration until the first cigarette remained significant (OR = 0.9, $p < 0.001$). A downward trend was observed in all values except TG, LDL, insulin, HOMA-Ir, and SBP from normal to moderate to high addiction levels. However, in the multivariate analysis, none of the variables remained statistically significant.

In Table 4, each anxiety type is divided into two groups, "Normal to Moderate" and "Severe". State and trait anxiety scores were elevated in 52.4% and 52.9% of the entire sample, respectively.

Across the spectrum from normal to severe state anxiety levels, there was a tendency for all values to increase, while the time to the first cigarette smoked tended to decrease. State anxiety was associated with differences in baseline glycemia, with the 'Severe' group exhibiting higher levels ($p = 0.050$). Additionally, state anxiety showed significant

**Table 3 Association between variables and metabolic syndrome in relation to nicotine dependence.**

| Variable | Normal to moderate | High | p | Multivariate analysis | |
| --- | --- | --- | --- | --- | --- |
| | (n = 42) | (n = 166) | | OR (95% CI) | p |
| Age (years) | 47.9 ± 13.2 | 42.8 ± 11.2 | 0.012 | 1.0 [0.9–1.0] | 0.299 |
| First cigarette (min) | 22.6 ± 9.2 | 9.9 ± 8.2 | 0.000 | 0.9 [0.8–0.9] | 0.000 |
| Years smoked (years) | 25.9 ± 11.7 | 22.6 ± 10.3 | 0.079 | 1.0 [0.9–1.0] | 0.717 |
| Basal glycemia (mg/dL) | 96.9 ± 17.2 | 95.8 ± 18.9 | 0.739 | | |
| TG (mg/dL) | 153.3 ± 76.0 | 159.0 ± 91.0 | 0.709 | | |
| HDL Col (mg/dL) | 50.9 ± 11.4 | 49.5 ± 16.3 | 0.606 | | |
| LDL Col (mg/dL) | 133.3 ± 41.0 | 124.6 ± 34.3 | 0.161 | 1.0 [1.0–1.0] | 0.421 |
| Insulin | 13.9 ± 13.9 | 11.7 ± 8.7 | 0.196 | 1.0 [0.9–1.0] | 0.453 |
| HOMA-IR | 3.7 ± 5.0 | 3.0 ± 3.2 | 0.260 | | |
| HbA1c | 5.7 ± 0.7 | 5.7 ± 0.8 | 0.946 | | |
| SBP (mm/hg) | 118.2 ± 16.2 | 117.8 ± 14.5 | 0.882 | | |
| DBP (mm/hg) | 73.9 ± 11.8 | 74.7 ± 11.6 | 0.701 | | |
| BMI | 26.1 ± 4.4 | 26.7 ± 5.0 | 0.507 | | |
| State anxiety | 48.5 ± 5.4 | 49.6 ± 6.9 | 0.324 | | |
| Trait anxiety | 48.1 ± 5.5 | 50.5 ± 7.2 | 0.050 | 1.0 [1.0–1.1] | 0.147 |

Note:
[a] Patients were divided into two categories based on the addiction scores calculated using the dependency scale.

**Table 4 Results stratified by STAI-S and STAI-T scores in two groups.**

| Variable | State anxiety[a] | | | Trait anxiety[b] | | |
| --- | --- | --- | --- | --- | --- | --- |
| | Normal to moderete n = 99 (47.6%) | Severe n = 109 (52.4%) | p | Normal to moderete n = 98 (47.1%) | Severe n = 110 (52.9%) | p |
| Age (years) | 42.2 ± 11.4 | 45.3 ± 12.0 | 0.058 | 41.8 ± 12.0 | 45.6 ± 11.4 | 0.019 |
| First cigarette (min) | 12.8 ± 9.6 | 12.2 ± 10.1 | 0.653 | 14.1 ± 10.2 | 11.1 ± 9.2 | 0.028 |
| Years smoked (years) | 22.0 ± 10.2 | 24.4 ± 11.0 | 0.097 | 22.0 ± 11.5 | 24.4 ± 9.8 | 0.098 |
| Basal glycemia (mg/dL) | 93.4 ± 12.8 | 98.3 ± 22.4 | 0.050 | 90.9 ± 10.6 | 100.5 ± 22.6 | 0.000 |
| TG (mg/dL) | 154.6 ± 85.7 | 161.0 ± 90.4 | 0.604 | 135.9 ± 73.9 | 177.5 ± 95.1 | 0.000 |
| HDL Col (mg/dL) | 49.3 ± 11.3 | 50.2 ± 18.4 | 0.664 | 53.3 ± 18.9 | 46.7 ± 10.6 | 0.003 |
| LDL Col (mg/dL) | 126.0 ± 37.5 | 126.8 ± 34.4 | 0.868 | 125.2 ± 34.2 | 127.5 ± 37.4 | 0.644 |
| Insulin | 11.4 ± 10.2 | 12.8 ± 9.8 | 0.339 | 10.4 ± 9.5 | 13.7 ± 10.2 | 0.015 |
| HOMA-IR | 2.8 ± 3.6 | 3.4 ± 3.7 | 0.260 | 2.5 ± 3.3 | 3.7 ± 3.8 | 0.013 |
| HbA1c | 5.6 ± 0.5 | 5.8 ± 0.9 | 0.034 | 5.4 ± 0.4 | 6.0 ± 0.9 | 0.000 |
| SBP (mm/hg) | 115.6 ± 12.5 | 120.0 ± 16.3 | 0.029 | 114.1 ± 11.8 | 121.2 ± 16.4 | 0.000 |
| DBP (mm/hg) | 73.5 ± 10.5 | 75.4 ± 12.5 | 0.233 | 72.2 ± 10.5 | 76.6 ± 12.1 | 0.005 |
| BMI | 26.2 ± 5.0 | 27.0 ± 4.8 | 0.277 | 25.3 ± 3.7 | 27.7 ± 5.5 | 0.000 |
| State anxiety points | 44.8 ± 4.5 | 53.6 ± 5.2 | 0.000 | 47.0 ± 5.9 | 51.6 ± 6.5 | 0.000 |
| Trait anxiety points | 47.7 ± 5.7 | 52.1 ± 7.4 | 0.000 | 44.6 ± 3.3 | 54.8 ± 5.7 | 0.000 |

Note:
[a,b] Patients were divided into two groups based on their STAI (State-Trait Anxiety Inventory) scale scores.

differences in HbA1c levels, with higher levels in the 'Severe' group ($p = 0.034$). Systolic blood pressure was significantly higher in the 'Severe' group for state anxiety, while diastolic blood pressure did not differ ($p = 0.029$; $p = 0.233$). State anxiety was significantly associated with trait anxiety, with higher values observed in the 'Severe' group ($p = 0.000$). Regarding anxiety scores across state anxiety subgroups, the mean state anxiety scores were 44.8 ± 4.5 and 53.6 ± 5.2, while the mean trait anxiety scores were 47.7 ± 5.7 and 52.1 ± 7.4 ($p = 0.000$).

State anxiety increased with the severity of anxiety, while the time to the first cigarette smoked tended to decrease. Baseline glycemia also showed a significant relationship with State anxiety. Specifically, the severe anxiety group exhibited higher baseline glycemia levels ($p = 0.05$) compared to other groups. In addition to baseline glycemia, state anxiety was associated with HbA1c levels. The severe anxiety group exhibited significantly higher HbA1c levels compared to the other groups ($p = 0.034$). SBP was significantly higher in the severe anxiety group compared to other groups ($p = 0.029$), whereas DBP did not show significant differences across anxiety levels ($p = 0.233$). State anxiety was significantly associated with trait anxiety, with higher trait anxiety scores observed in the severe anxiety group ($p = 0.000$). Regarding anxiety scores across state anxiety subgroups, the mean state anxiety scores were 44.8 ± 4.5 and 53.6 ± 5.2, while the mean trait anxiety scores were 47.7 ± 5.7 and 52.1 ± 7.4. These differences were statistically significant ($p = 0.000$).

## DISCUSSION

The etiology of the metabolic syndrome remains subject to debate, yet smoking emerges as a notable modifiable risk factor. Extensive research indicates that smoking is linked to lipid irregularities, endothelial dysfunction, and a prothrombotic state, all of which constitute components of metabolic syndrome (*Golbidi, Edvinsson & Laher, 2020*; *Slagter et al., 2013*). This association is further underscored by evidence suggesting that smoking may exacerbate insulin resistance, contributing to metabolic and hemodynamic aberrations (*Slagter et al., 2013*). While certain investigations have demonstrated a positive correlation between smoking and the prevalence of metabolic syndrome (*Calo et al., 2013*; *Chen et al., 2008*; *Shin, Oh & Cho, 2018*; *Wada, Urashima & Fukumoto, 2007*), conflicting findings have been reported in other studies (*Katano et al., 2010*; *Yu et al., 2014*). Notably, a study among Turkish women suggested a reduced risk of metabolic syndrome among smokers (*Onat et al., 2007*). While previous studies have typically focused on daily tobacco consumption or cumulative smoking duration, our study offers a nuanced analysis incorporating gender-specific median values, pack years, a three-tier assessment (light, moderate, heavy), and quartiles, aligning with prior research (*Cena et al., 2013*; *Wada, Urashima & Fukumoto, 2007*; *Zhu et al., 2011*). Additionally, our study unveils, for the first time, parallels between key metabolic components and anxiety scores within a sample of 208 smokers.

In exploring the relationship between anxiety and metabolic syndrome, evidence suggests a tenuous link (*Butnoriene et al., 2014*; *Skilton et al., 2007*). A recent meta-analysis of 18 cross-sectional studies examining anxiety's impact on metabolic syndrome risk reported a modest yet discernible increase in risk (OR = 1.07) (*Tang, Wang & Lian, 2017*).

Furthermore, a registry review revealed elevated cardiometabolic risks among individuals with anxiety, encompassing conditions such as diabetes, hypertension, hyperlipidemia, and obesity (*Pérez-Piñar et al., 2017*). Given the profound influence of lifestyle on metabolic syndrome and the generally unhealthy lifestyles prevalent among psychiatric patients, it is posited that poor lifestyle choices may contribute to the observed association between anxiety disorders and metabolic syndrome (*Penninx & Lange, 2022*).

Nicotine, carbon monoxide, and other smoking metabolites have been implicated in inducing insulin resistance. *Cena et al. (2013)* have noted that the effects previously attributed to insulin resistance persist even when considering smoking effects independently of insulin resistance. The absence of dose-dependent worsening in insulin and HOMA-IR levels in our study compared to the existing literature is a notable and distinctive finding. Despite this, we observed that criteria for metabolic syndrome and important metabolic components were affected at various levels, including SBP, DBP, HbA1c, and LDL. Consequently, mechanisms other than insulin resistance can adversely influence metabolic syndrome parameters, revealing an ongoing deterioration independent of insulin resistance, just as suggested by *Cena et al. (2013)*.

A comparative analysis of men and women revealed no significant differences in BMI and age. However, as anticipated, disparities were observed in HDL values, with men exhibiting higher SBP and DBP levels and a higher prevalence of poor triglyceride profiles. Another notable difference was the higher proportion of heavy smokers among men; this may be evidence of a clear relationship between the smoking profile and SBP, DBP and triglyceride profiles. This observation is supported by our study sample, which adjusts for confounding variables such as BMI, age, metabolic syndrome-related diseases, or psychiatric illness status. Previous research by *Chen et al. (2008)* found no significant relationship between fasting blood glucose levels and tobacco consumption. At the same time, conflicting evidence was reported by *Will et al. (2001)*, who suggested an association between increasing tobacco consumption and the incidence of type 2 diabetes (*Chen et al., 2008*). These contradictory findings regarding the effect of smoking on fasting blood glucose levels underscore the complexity of this relationship. Thus, while our study did not reveal a distinct difference in fasting blood glucose, definitive conclusions cannot be drawn.

Our study also evidently impacted LDL profile, an important risk factor for coronary artery disease, although LDL is not among the criteria for metabolic syndrome. Specifically, LDL profile was lower in light smokers or those with less than 10 years of smoking history. However, the duration of this effect could not be determined statistically in our study. While triglyceride levels increased significantly in accordance with the median smoking duration (25 years), the duration of this effect remained unclear statistically like LDL profile. This ambiguity may be attributed to the cross-sectional nature of our study design. As anticipated, SBP and DBP profiles were significantly impaired in heavy smokers, consistent with findings in the literature (*Cena et al., 2013*).

Research suggests that trait anxiety is more likely to be associated with metabolic syndrome markers, as persistent anxiety could lead to sustained stress responses that impact physiological processes, including metabolic pathways. Research highlights that

trait anxiety could exacerbate the negative effects of smoking on metabolic health, as individuals with high trait anxiety may use smoking as a coping mechanism, which intensifies nicotine dependence and metabolic syndrome risk factors (*Chida & Steptoe, 2009*; *Cohen, Edmondson & Kronish, 2015*). A study by *McEwen (1998)* demonstrated that individuals with high trait anxiety tend to have higher levels of smoking-related biomarkers, such as cortisol and catecholamines, potentially worsening lipid profiles and blood pressure. Consistent with previous research, our findings suggest that SBP and DBP values are influenced by the anxiety levels of patients, as discussed elsewhere in the literature (*Lemche, Chaban & Lemche, 2016*; *Marvar et al., 2014*). Furthermore, it has been established in the literature that a poor metabolic profile can lead to anxiety (*Butnoriene et al., 2014*; *Diamanti-Kandarakis, 2008*; *Diamanti-Kandarakis et al., 1999*). Given these findings, it is evident that individuals seeking smoking cessation, particularly those with a compromised metabolic profile, may need integrated services addressing both physical and mental health concerns, including anxiety.

Anxiety levels and blood pressure have been found to be related in our study. Potential mechanisms explaining this relationship could involve physiological stress responses. Anxiety often activates the sympathetic nervous system, leading to the release of stress hormones such as cortisol and adrenaline, which can increase heart rate and cause vasoconstriction, ultimately raising blood pressure (*Chrousos, 2009*). Chronic anxiety may result in sustained sympathetic activation, contributing to long-term increases in blood pressure (*Wenner, 2018*). Moreover, behavioral factors could also play a role. Individuals with heightened anxiety levels may be more likely to engage in unhealthy behaviors, such as poor dietary choices, smoking, or lack of exercise, all of which are risk factors for hypertension (*Hering, Lachowska & Schlaich, 2015*; *King et al., 1996*). Furthermore, anxiety may influence the autonomic nervous system and the regulation of blood pressure through altered baroreceptor sensitivity (*Ziegler, 2012*).

Individuals with a history of smoking are reported to be more susceptible to metabolic syndrome than non-smokers, with a significant risk across age groups and, as in our study, across genders (*Ford, Li & Zhao, 2010*; *Grundy et al., 2005*). Dietary habits, physical activity levels, and socioeconomic status are also important factors influencing metabolic differences (*Hildrum et al., 2007*; *Lakka & Laaksonen, 2007*; *Matthews et al., 2008*; *Santos-Marcos, Perez-Jimenez & Camargo, 2019*). The majority of our sample is in stages just before the age groups most relevant for metabolic syndrome. According to reports, age may partially influence the prevalence of metabolic syndrome, which increases significantly with age. However, given that the entire sample was cross-sectional and collected over a 1-year period from a specific center, we assume that these factors did not significantly influence the study outcomes. The uniformity in the sample, along with uninterrupted data collection during this period, helps minimize variability due to these confounders. Furthermore, the specific center's population characteristics were consistent, which we believe reduces the potential impact of socioeconomic status, diet, and physical activity on the findings.

Research on post-cessation recovery reveals that the timeline and degree of metabolic improvements are often proportional to pre-cessation tobacco use, with studies observing

a gradual normalization of lipid, glucose, and blood pressure values up to 5–20 years post-cessation, depending on baseline smoking intensity (*Chiolero et al., 2008*; *Park et al., 2021*; *Wada, Urashima & Fukumoto, 2007*; *Will et al., 2001*; *Zhu et al., 2011*). For instance, a study demonstrated that those with heavier smoking histories took longer to exhibit improvements in glucose metabolism and insulin resistance (*Wada, Urashima & Fukumoto, 2007*). Significant improvements were observed one month post-cessation: respiratory function markers such as FEV1 and FEF25/75 improved, eCO levels dropped substantially, and respiratory symptoms alleviated. Metabolic parameters also showed positive changes, including a modest increase in vitamin D levels (without supplementation) and reductions in total cholesterol (*Pezzuto et al., 2023*).

In terms of mental health, incorporating follow-up data could further elucidate whether anxiety decreases alongside metabolic improvements post-cessation. While our study suggests elevated anxiety in heavy smokers, studies indicate that cessation may reduce anxiety and improve quality of life, likely due to metabolic and neurochemical stabilization (*Hajek, Taylor & McRobbie, 2010*; *Zvolensky et al., 2018*). As anxiety and smoking cessation are intertwined, addressing metabolic health post-cessation may thus provide dual benefits for smokers. Besides our current study was limited to specific psychological metrics, we agree that incorporating additional nuanced psychological measurements, such as stress resilience or coping mechanisms, could provide a more comprehensive understanding of the observed trends.

## Limitations

While our study offers valuable insights into the interplay between smoking, anxiety, addiction levels, and metabolic syndrome components, it has some limitations. By acknowledging these limitations, we aim to provide a balanced interpretation of our findings and emphasize the need for further research to build on our results.

**1. Measurement gaps:**

The absence of waist circumference measurements and data on the prevalence of metabolic syndrome is a notable limitation. Waist circumference is a critical component in defining metabolic syndrome and provides valuable insights into obesity-related risks. Future research should include these measurements to align with established definitions and enhance the accuracy of metabolic syndrome assessments.

**2. Uncontrolled confounding factors:**

We did not account for several lifestyle factors, such as alcohol consumption, dietary habits, and physical activity levels, which are known to influence both metabolic syndrome and anxiety. These unmeasured confounders may have impacted our findings. Future studies should incorporate these variables to provide a more comprehensive understanding of the relationships explored.

**3. Selection bias**

First, the inclusion criteria excluded individuals with chronic health conditions and a BMI over 30, which may have led to the underrepresentation of smokers with comorbidities often associated with tobacco use. Second, the study was conducted at a single outpatient

smoking cessation clinic, which may limit the applicability of findings to less motivated smokers to quit.

**4. Potential bias in self-reported data:**

The reliance on self-reported smoking history, particularly for estimating pack years, introduces the possibility of recall bias. This limitation may affect the accuracy of the smoking history, which is crucial for assessing smoking intensity and related health outcomes.

**5. Cross-sectional design constraints**

a) Limited to the relevant section about associations between anxiety scores, components of metabolic syndrome, and smoking intensity, the cross-sectional nature of our data means we cannot determine the directionality of these relationships.

b) The temporal sequence of events cannot be determined for identified associations. Future research employing longitudinal designs would be beneficial to address this constraint and provide deeper insights.

# CONCLUSIONS

In conclusion, our study underscores the interconnectedness of nicotine dependence and anxiety levels with metabolic syndrome parameters among smokers. Metabolic syndrome components—including systolic and diastolic blood pressure, HbA1c, LDL, and triglycerides—show an increase in individuals who smoke, with distinct differences between light and heavy smokers. Notably, systolic blood pressure exhibits a dose-response relationship with smoking intensity, suggesting that heavier smoking may further elevate this risk factor. Additionally, there is an observed link between anxiety levels and blood pressure, emphasizing a potential interplay between psychological stress and cardiovascular risk in smokers. These findings highlight the critical link between physical and mental health in individuals who smoke. Moving forward, it is imperative to conduct further research in this area and to integrate mental health considerations into smoking cessation programs. We suggest that future research employs more detailed psychiatric assessments to better understand the psychological impact of smoking. By addressing both physical and mental health aspects, we can better support individuals in their efforts to quit smoking and improve overall well-being.

## Perspective

1) With the absence of a clear dose-dependent pattern in insulin and HOMA-IR levels, this research supports the studies point to potentially alternative mechanisms at play.

2) Such studies would provide valuable insights into the timeline of worsening with smoking and extent of recovery following cessation, particularly for individuals with a history of heavy smoking.

3) By bridging the gaps between physical and psychological health, this study sets the stage for more effective interventions that target both the root causes and long-term consequences of smoking.

4) Longitudinal investigations are also necessary to better understand the progression of these relationships and to determine whether metabolic improvements post-smoking cessation correlate with reductions in anxiety.

These results underscore the need for extended post-cessation monitoring and the development of targeted strategies to address residual metabolic risks.

### Funding
The authors received no funding for this work.

### Competing Interests
The authors declare that they have no competing interests.

### Author Contributions
- Musa Şahin conceived and designed the experiments, performed the experiments, analyzed the data, prepared figures and/or tables, authored or reviewed drafts of the article, and approved the final draft.
- Didem Yüzügüllü conceived and designed the experiments, performed the experiments, authored or reviewed drafts of the article, and approved the final draft.

### Human Ethics
The following information was supplied relating to ethical approvals (*i.e.*, approving body and any reference numbers):

Ethical clearance for the study was obtained from Çukurova University on 08.04.2022, with acceptance number 22, and informed consent was obtained from all participants (Ethical Application: 121/22, Date: 08.04.2022).

### Ethics
The following information was supplied relating to ethical approvals (*i.e.*, approving body and any reference numbers):

Ethical approvals obtained from Çukurova University in adherence to Helsinki Declaration principles (Ethical Application: 121/22, Date: 08.04.2022).

### Data Availability
The data is available at figshare: SAHIN, Musa (2024). Metabolic Syndrome and Anxiety Levels in Light and Heavy Smokers. figshare. Dataset. https://doi.org/10.6084/m9.figshare.25750842.v1.

## Supplemental Information

Supplemental information for this article can be found online at http://dx.doi.org/10.7717/peerj.19069#supplemental-information.

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
