# Peer review of "Outcomes of metabolic syndrome and anxiety levels in light and heavy smokers"

_PeerJ, doi:10.7717/peerj.19069_

## Round 0.1 · original submission · Major Revisions

Dear Dr. Şahin,

Your manuscript entitled “Outcomes of metabolic syndrome and anxiety levels in light and heavy smokers", which you submitted to PeerJ, has been reviewed by the editor and 3 external reviewers.

I regret to inform you that the reviewers have raised significant concerns that need to be addressed before the manuscript can be further considered. Specifically, the study's limitations and analysis methods should be more thoroughly addressed and explained. However, since reviewers find some merit in the paper, I would be willing to reconsider if you wish to undertake major revisions and resubmit.

If you decide to resubmit the revised version, please summarize all the improvements made in the new version and give answers to all critical points raised in the reviewers’ report in an accompanying letter. Copy and paste each and every reviewer's comment above your response.

Please note that resubmitting your manuscript does not guarantee eventual acceptance. Since the requested changes are major, the revised manuscript will undergo a second round of review by the same reviewers. I must emphasize that the acceptability of the revision will depend upon the resolution of the points raised by the reviewers.

Sincerely yours,

Stefano Menini

·

Basic reporting

1. Abstract: adequate structure related to the paper.
2. Introduction: It contains main and relevant ideas current on the subject being analyzed.
3. "Materials and Methods" and "Discussion" chapters are well presented and well balanced.

Experimental design

Methods are well detailed and analyzed parameters are consistent with the topic. The limits of
the study are also mentioned.

Validity of the findings

The analyzed subject is of the interest to multiple medical specialties: general medicine, internal medicine, cardiology, diabetes and nutritional diseases , psychiatry and psychology. I support the publication of these types of articles because they contribute to the accumulation of data in an area that affects a significant number of patients and involves complex and long-term medical care. The scientific value of the article would increase significantly by restructuring of " Conclusions" chapter
which should reflect the results of the study, general discussions and opinions of the authors.

Reviewer 2 ·

Basic reporting

on line 27 the sentence: While anxiety levels did not exhibit a
significant correlation with smoking status, this is not strenghthened in the abstract conclusions.
Please explain in the introduction that such dependence is covered in the ICD-11 classification of diseases.

Some missing references I suggest to include to improve the discussion, about the efficacy of smoking cessation program on metabolism and respiratory system, the association of mood disorders and smoking dependence and the evaluation of patient expectations

Experimental design

it should be pointed out that STAI is a 40 self-report items questionnaire
it needs to report in more detail on the inclusion criteria
Is it a retrospective analysis?
please explain the laboratory method for detecting and assaying metabolites
was a motivational questionnaire administered?

Validity of the findings

the findings are valid and easy to read
however, please report the main comorbidities
please state if all patients were current smokers
in table 3 provide further information about the cutoff value to distinguish patients with moderate from patients with high addiction value
is it possible a multivariate analysis about probability of smoking cessation?

·

Basic reporting

See below

Experimental design

See below

Validity of the findings

See below

Additional comments

Positive Points

1. Novelty and Relevance:
- The study addresses an important intersection of public health issues: smoking, metabolic syndrome, and anxiety. These are highly relevant topics as they pertain to both physical and mental health, especially in the context of smoking cessation. The examination of metabolic syndrome (MS) in relation to nicotine addiction and anxiety levels provides valuable insights into holistic health management.
- The use of pack-years (PY) as a metric to categorize smoking into light, moderate, and heavy smoking groups adds a layer of precision and granularity, allowing for a clearer understanding of the dose-dependent impact of smoking on metabolic and psychological health.

2. Study Design and Methodology:
- The cross-sectional study design is appropriate for the initial investigation of correlations between metabolic syndrome parameters and anxiety levels.
- The inclusion of biochemical parameters such as hemoglobin A1c (HgbA1C), low-density lipoprotein (LDL), and triglycerides (TG) is commendable as these are clinically relevant markers of metabolic health. The addition of anxiety assessment through the State-Trait Anxiety Inventory (STAI) further strengthens the study's holistic approach.
- The study adheres to ethical standards, with approval from Çukurova University and adherence to the Declaration of Helsinki. This is an essential aspect of any clinical research.

3. Statistical Rigor:
- The sample size determination using the Epi Info Sample Size Calculator and the 95% confidence interval indicates a well-planned statistical approach. A p-value threshold of 0.05 for statistical significance is also appropriate.

4. Findings and Insights:
- The study’s findings regarding the dose-dependent relationship between smoking status and key metabolic parameters (e.g., SBP, DBP, HgbA1c, LDL) are clinically significant. These results contribute to the understanding of how smoking exacerbates metabolic syndrome.
- The observation that anxiety levels tended to increase with higher SBP and DBP values adds an interesting psychological dimension to the physical health outcomes.
* * *
Negative Points

1. Lack of Clarity in Results Presentation:
- While the study provides a wealth of data, the presentation of results is somewhat cluttered and lacks clarity. For instance, in Table 1, the comparison between genders is mentioned, but the exact values and their statistical significance are not clearly outlined in the text. The results section should be restructured to provide more explicit interpretation of the tables and figures. For example, it could be helpful to precisely state: "Males exhibited a significantly higher proportion of heavy smokers (p = 0.008) and individuals meeting the TG criteria for metabolic syndrome (p = 0.001)."

2. Absence of Waist Circumference Data:
- One of the key components of metabolic syndrome is abdominal obesity, often measured by waist circumference. The omission of this crucial parameter is a significant limitation. Although the study measures BMI, this is not always a reliable indicator of visceral fat. Future studies should include waist circumference to provide a more comprehensive assessment of metabolic syndrome.

3. Cross-Sectional Design Limitations:
- While a cross-sectional design is appropriate for investigating correlations, it limits the ability to infer causality. The authors should acknowledge this limitation more explicitly in the discussion and suggest the need for longitudinal studies to better assess the causal relationships between smoking, metabolic syndrome, and anxiety.

4. Inconsistent Findings on Anxiety and Smoking:
- The study notes that anxiety levels did not show a statistically significant correlation with smoking status, yet there is a trend of increased anxiety with higher SBP/DBP. This contradiction warrants further investigation. The authors should explore this inconsistency more deeply in the discussion, offering potential explanations for the lack of a stronger correlation. For example, they could hypothesize that other factors, such as social or environmental stressors, may play a more significant role in anxiety levels than smoking alone.

5. Limited Discussion of Confounding Variables:
- Although the study excludes individuals with diagnosed psychiatric illnesses, it does not account for potential confounding variables such as alcohol consumption, physical activity levels, or dietary habits, all of which could influence both metabolic syndrome and anxiety. The authors should either control for these confounders or at least discuss their potential impact in the limitations section.

6. Insufficient Exploration of Heavy Smokers Post-Cessation Metabolic Profiles:
- The article mentions that the metabolic profile may persist even after smoking cessation, but it does not provide sufficient data or exploration of this issue. Including a follow-up on smokers who have quit could enhance the study’s contribution to understanding how quickly metabolic syndrome parameters recover after cessation.
* * *
Recommendations for Improvement

1. Provide Clearer Interpretation of Results:
- The results section needs clearer narratives that correlate with the data in the tables. For example, rather than just stating “significant differences were observed across all three categories” (referring to light, moderate, and heavy smokers), the authors should provide more detail: "Light smokers exhibited significantly lower LDL levels compared to moderate and heavy smokers (p = x), suggesting a potential dose-dependent protective effect against dyslipidemia."

2. Incorporate Waist Circumference as a Metabolic Syndrome Criterion:
- Future studies should include waist circumference measurements to provide a complete assessment of metabolic syndrome. This would align the study with established definitions of metabolic syndrome and offer a more accurate evaluation of obesity-related risks.

3. Acknowledge the Cross-Sectional Nature More Explicitly:
- The authors should emphasize the limitations of a cross-sectional design more explicitly in the discussion. They should recommend that longitudinal studies be conducted to assess whether smoking cessation leads to improvements in metabolic syndrome parameters and anxiety levels over time.

4. Explore Confounding Variables:
- Future iterations of this study should control for alcohol consumption, dietary habits, and physical activity levels. These variables could significantly impact both metabolic syndrome and anxiety and should be accounted for in the analysis.

5. Elaborate on the Anxiety Findings:
- The authors should provide a more in-depth discussion of the inconsistent findings on anxiety. For instance, they could explore whether the type of anxiety (state vs. trait) might influence the relationship between smoking and metabolic syndrome. They could also suggest that future research employs more detailed psychiatric assessments to better understand the psychological impact of smoking.

6. Include Smoking Cessation Outcomes:
- The article would benefit from a follow-up study that tracks the metabolic recovery of heavy smokers after quitting. This could help clarify whether metabolic syndrome components improve in a dose-dependent manner after cessation and whether anxiety levels decrease as metabolic health improves.
* * *
Conclusion

In conclusion, this study makes a valuable contribution to the literature by exploring the relationship between smoking status, metabolic syndrome, and anxiety. However, there are several areas where the study could be improved, particularly in terms of clarity of results presentation, inclusion of waist circumference data, and control of confounding variables. Addressing these limitations would enhance the study’s rigor and provide more actionable insights for clinical practice, particularly in the context of smoking cessation programs. The authors are encouraged to build upon these findings with more detailed analyses and longitudinal follow-ups to further elucidate the complex interplay between smoking, metabolic health, and psychological well-being.

---

## Round 0.2 · Major Revisions

Dear Dr. Şahin,

Thank you for your resubmission. I have received the report from the reviewers, who have suggested substantial modifications. I urge you to pay careful attention to these points and invite you to address the reviewers' comments and recommendations.

I hope you will be prepared to make the necessary amendments and submit a revised manuscript with a statement of how you have responded to the criticisms raised. Please copy and paste each and every reviewer's comment above your response. You are also kindly requested to provide a complete tracked changes version of the manuscript to make verifying that the required changes have been made easier.

I look forward to receiving your revision,

Sincerely yours,

Stefano Menini

Reviewer 2 ·

Basic reporting

it is interesting and clear

Experimental design

aims are defined and clear, please
greater clarity is needed in the inclusion criteria

Validity of the findings

the impact and novelty are good
I suggest to include the following manuscript as a reference since the manuscript deals with nicotine addiction and associated metabolic and functional factors
-Int J Chron Obstruct Pulmon Dis. 2023 Dec 1;18:2861-2865.

Additional comments

I suggest to include a perspective section

·

Basic reporting

Negative Points:

The results section could benefit from better organization and clarity in presenting the findings.
Some tables lack clear titles or descriptions, making it difficult to understand the data presented.
Recommendations:

Consider restructuring the results section to present the findings in a more logical and coherent manner, possibly by grouping related outcomes together.
Provide clear and descriptive titles for all tables, ensuring that the reader can easily understand the data being presented.

Experimental design

Negative Points:

The retrospective nature of the study and the reliance on self-reported information may introduce recall bias and inaccuracies.
The study lacks a control group of non-smokers, which could provide valuable insights into the differences between smokers and non-smokers in terms of metabolic syndrome parameters and anxiety levels.
Recommendations:

Consider incorporating a prospective component to the study design, which could reduce the potential for recall bias and allow for more accurate data collection.
Include a control group of non-smokers to enable direct comparisons and better understand the impact of smoking on metabolic syndrome and anxiety.

Validity of the findings

Negative Points:

The cross-sectional nature of the study precludes the establishment of causal relationships between the variables under investigation.
The study sample is drawn from a single smoking cessation clinic, which may limit the generalizability of the findings to the broader population.
Recommendations:

Clearly acknowledge the limitations of the cross-sectional design and the inability to infer causality from the observed associations.
Consider conducting a multi-center study or recruiting participants from diverse settings to enhance the generalizability of the findings.

Additional comments

Negative Points:

The discussion section could be more concise and focused, as it tends to be repetitive in some parts.
The study lacks a discussion of potential confounding factors or effect modifiers that may influence the observed associations.
Recommendations:

Streamline the discussion section by focusing on the most salient findings and their implications, while avoiding unnecessary repetition.
Discuss potential confounding factors or effect modifiers, such as dietary habits, physical activity levels, or socioeconomic status, and how they may have influenced the study results.
Consider including a limitations section to explicitly acknowledge the study's limitations and potential sources of bias.
Overall, the study provides valuable insights into the relationships between smoking status, metabolic syndrome parameters, and anxiety levels. However, addressing the identified limitations and incorporating the recommended improvements could enhance the validity and impact of the findings.

---

## Round 0.3 · Minor Revisions

Dear Dr. Şahin,

Thank you for your resubmission. Below, I have included the comments from Reviewer 3, emphasizing additional points that need your careful attention before we can further consider your submission. I strongly encourage you to address each point comprehensively.

Please copy and paste each reviewer's comment above your corresponding response when revising your manuscript. Additionally, please provide a complete version of the manuscript with tracked changes to facilitate the verification of the revisions made.

Please be aware that resubmitting your manuscript does not guarantee acceptance. The acceptability of the revised submission will depend on the satisfactory resolution of all issues raised by the reviewers.

Sincerely yours,

Stefano Menini

·

Basic reporting

"Outcomes of Metabolic Syndrome and Anxiety Levels in Light and Heavy Smokers"

Strengths of the Manuscript:

1. Comprehensive Research Design
- Robust methodology with clear inclusion/exclusion criteria
- Utilized standardized assessment tools (Fagerstrom Test, State-Trait Anxiety Inventory)
- Comprehensive data collection across multiple metabolic parameters
- Detailed statistical approach with multiple analytical strategies

2. Methodological Rigor
- Ethical clearance obtained from Çukurova University
- Well-defined sampling strategy
- Precise measurement techniques for biochemical parameters
- Transparent reporting of sample characteristics

Weaknesses and Recommendations:

1. Methodological Clarifications
Recommendation: Provide more detailed explanation of:
- Precise recruitment process
- Potential selection biases
- Criteria for excluding participants with pre-existing conditions

Example improvement: "While our exclusion criteria eliminated potential confounding factors, future studies could explore how these conditions interact with smoking status."

2. Statistical Analysis
Potential Enhancement:
- Consider more advanced multivariate modeling
- Provide confidence intervals alongside statistical significance
- Elaborate on the rationale for specific statistical tests used

Example suggestion: "Supplementary analysis using advanced regression techniques could further elucidate the nuanced relationships between smoking status and metabolic parameters."

3. Anxiety Correlation
Recommendation:
- Deeper exploration of non-significant anxiety correlations
- Discuss potential mechanisms explaining observed trends
- Consider additional psychological measurements

Example improvement: "While our current analysis did not reveal significant anxiety correlations, future research could investigate more nuanced psychological metrics."

4. Language and Presentation
Recommendations:
- Standardize terminology (e.g., consistent use of "pack-years")
- Minimize typographical errors
- Enhance clarity of statistical interpretations

5. Limitations Discussion
Recommendation: Explicitly acknowledge study limitations
- Cross-sectional design constraints
- Potential regional specificity
- Self-reported smoking history limitations

Positive Concluding Observations:
- Significant contribution to understanding smoking's metabolic impacts
- Comprehensive approach to investigating complex health interactions
- Clear potential for informing smoking cessation strategies

Overall Assessment:
The manuscript presents a valuable scientific contribution with promising insights into smoking's metabolic effects. With targeted refinements, it has strong potential for publication.

Recommendation: Major Revision

Specific Recommendations for Improvement:
1. Enhance statistical analysis depth
2. Provide more comprehensive limitation discussion
3. Clarify methodological nuances
4. Refine language precision

Suggested Action: Revise manuscript addressing aforementioned points before resubmission.

Experimental design

revise

Validity of the findings

revise

Additional comments

revise

---

## Round 0.4 · accepted · Accept

Dear Dr. Şahin,

Thank you for submitting the revised version of your manuscript. After a thorough review of the changes by the reviewers and myself, I am pleased to inform you that all the reviewers' comments have been adequately addressed. Therefore, your manuscript is ready for publication in PeerJ.

I thank all reviewers for their efforts in improving the manuscript and the authors' cooperation throughout the review process.

Sincerely yours,
Stefano Menini

·

Basic reporting

the article is good

Experimental design

the article is good

Validity of the findings

the article is good

Additional comments

the article is good